# Electrodeposition of Bi from Choline Chloride-Malonic Acid Deep Eutectic Solvent

**DOI:** 10.3390/ma16010415

**Published:** 2023-01-01

**Authors:** Xiaozhou Cao, Hao Wang, Tianrui Liu, Yuanyuan Shi, Xiangxin Xue

**Affiliations:** 1School of Metallurgy, Northeastern University, Shenyang 110819, China; 2Northern Engineering and Technology Corporation, MCC, Dalian 116600, China; 3Institute of Metal Research, Chinese Academy of Sciences, Shenyang 110016, China

**Keywords:** bismuth, deep eutectic solvent, electrodeposition, cyclic voltammetry

## Abstract

Deep eutectic solvent (DES) has been widely used in the field of metal electrodeposition as an economical and environmentally friendly green solvent. Metallic bismuth films were prepared by electrodeposition from choline chloride-malonic acid (ChCl-MA) deep eutectic solvent (DES) containing BiCl_3_. Fourier transform infrared (FTIR) spectroscopy and Raman spectroscopy were used to study the structure of ChCl-MA-BiCl_3_, and the results showed that Bi(III) was in the form of [BiCl_6_]^3−^ ions. The viscosity of ChCl-MA-BiCl_3_ ranges from 200 to 1200 mPa·s at temperatures from 363 K to 323 K. The conductivity of 0.01 M Bi(III) in ChCl–MA is 3.24 ms·cm^−1^ at 363 K. The electrochemical behavior and electrodeposition of Bi(III) in DES were investigated by cyclic voltammetry (CV) and chronoamperometry. The results showed that the electrodeposition reaction was a quasi-reversible reaction controlled by the diffusion and the nucleation of bismuth was a three-dimensional instantaneous nucleation. The diffusion coefficient of Bi(III) in ChCl-MA was 1.84 × 10^−9^ cm^2^·s^−1^. The electrodeposition product was observed by scanning electron microscopy (SEM), and the results showed that the deposition potential has a significant influence on the morphology of the bismuth film. X-ray photoelectron spectroscopy (XPS) shows that bismuth and bismuth oxides are present in the deposited film obtained by electrodeposition.

## 1. Introduction

Bismuth (Bi) is a silvery-white metal with remarkable electrical, physical and chemical properties such as high density, high melting point and low conductivity, especially unique electronic properties which are widely used for catalysts and energy materials. Bismuth-based nanosheets are known as highly efficient electrocatalysts for the CO_2_ reduction reaction [1]. Bismuth can form alloys with lower melting points with metals such as tin and antimony and can be applied as sliding bearing overlay material in internal combustion engines [2]. Bi−Co films could be applied in photoelectronic, thermoelectric and photoelectrochemical devices and solar selectors [3]. Bismuth is used in medical applications for its good antibacterial properties. It is also used for the fabrication of magnetic sensors based on positive magnetoresistance variations [4].

Electrodeposition is an effective method for the fabrication of Bi, and the morphologies and particle sizes of Bi ranging from micrometers to nanometers can be obtained by controlling the electrodeposition conditions. Osipovich et al. studied the electrochemical behavior of Bi(III) ions on the Te electrode in HNO_3_-Bi(NO_3_)_3_ aqueous solution [5], while hydrogen evolution is a problem in aqueous systems. Castillejo et al. used cyclic voltammetry and chronoamperometry to study the electrochemical properties of BiCl_3_ in the PbCl_2_-KCl molten salt system at 475 °C [6]. The higher temperature of electrolysis leads to high consumption of power, which is mainly due to the high liquidus temperature of the molten salt electrolyte [7]. Ionic liquids (ILs) composed of solvent-free molten salts have received extensive attention as electrolytes due to their low vapor pressure, wide electrochemical windows, and high chemical and thermal stability [8,9]. Pan et al. investigated the electrodeposition of Bi in the 1-methyl-3-butylimidazolium chloride-aluminum chloride (MBIC-AlCl_3_) ionic liquid and found Bi adatoms and AlCl_4_^−^ coadsorption in MBIC-AlCl_3_ [10]. However, the high cost of ILs limits their large-scale application. 

Deep eutectic solvents (DESs) composed of quaternary ammonium salts and hydrogen bond donors were proposed by Abbott et al. [11,12]; they have properties similar to ILs, but they are cheaper, have higher ionic conductivity and are moisture-stable [13]. Hsieh et al. used different working electrodes to study the electrochemical behavior and electrodeposition of Bi in the choline chloride-ethylene glycol deep eutectic solvent (ChCl-EG-Bi(NO_3_)_3_) [13]. The results showed that the nucleation mechanisms of Bi deposited on glassy carbon electrodes, platinum and nickel electrodes were different, with three-dimensional instantaneous nucleation on glassy carbon electrodes and three-dimensional progressive nucleation on platinum and nickel electrodes. Although the deposition of Bi in different electrolytes has been studied, the effects of the physicochemical properties (conductivity, viscosity) of DES and the structure of Bi(III) in DES on the electrochemical behavior and nucleation mechanism of bismuth in ChCl-MA still require more research and discussion.

In the present study, Bi was prepared by electrodeposition in ChCl-MA-BiCl_3_. FTIR and Raman spectroscopy were used to study the structure of Bi(III) in DES. The electrochemical behavior and nucleation mechanism of bismuth in DES were investigated by CV and chronoamperometry. The relationship of the viscosity, conductivity and temperature was studied to further describe the voltammetric behavior of bismuth in ChCl-MA. The effect of cathodic potential on the morphological characteristics of deposits is discussed.

## 2. Materials and Methods

### 2.1. Electrolyte Preparation

Malonic acid (99%; Sinopharm Chemical Reagent Co., Ltd., Shanghai, China) and choline-chloride (98%; Aladdin Reagent Co., Ltd., Los Angeles, CA, USA) were dried at 333 K and 353 K, respectively, for 24 h in a vacuum drying oven. ChCl and MA were mixed at a molar ratio of 1:1, and heated at 333 K until a colorless homogeneous liquid formed. Bismuth chloride (98%; Aladdin Reagent Co.) was then added to ChCl-MA DES and heated to dissolve to form the electrolyte.

### 2.2. Spectral Experiment

The structure of the ChCl-MA blank system and ChCl-MA-BiCl_3_ electrolyte was studied by FTIR spectrometry (Nicolet-380, Thermo Technology, Minden, NV, USA) and Raman spectrometry (LABRAM HR800, HORIBA, Irvine, CA, USA). The FTIR test used a potassium bromide (KBr) cell window, and the measured wavelength range was 4000–400 cm^−1^. The Raman spectra test used a He–Ne laser operating at 633 nm with a 5 s irradiation at 30 mW. Laser power was focused using Olympus 50 × objective on the samples, while the laser spot was 3 μm, the grating size was 600 grooves/mm and the wave number shift range was 200–4000 cm^−1^. FTIR and Raman spectroscopy tests were performed at room temperature.

### 2.3. Viscosity and Conductivity Experiments

A rotational viscometer (DV2T, China Tianjin Brook Technology Co., Ltd., Tianjin, China) with an accuracy of ±1.0% of range with displayed test data and a conductivity meter (DDS-12A, Shanghai Lida Instrument Factory, Shanghai, China) with an accuracy of 0.01 mS/cm were used to measure the viscosity and conductivity of ChCl-MA-BiCl_3_ at different temperatures and concentrations. The measurement was maintained for 10 min at each temperature to achieve thermal equilibrium of DES. Viscosity and conductivity were measured three times under the same conditions, and the average value was recorded. The conductivity meter was calibrated using 0.01 M KCl standard solution before the measurement of DES. 

### 2.4. Electrochemistry and Electrodeposition Experiments

Electrochemical experimental methods, including cyclic voltammetry (CV) and chronoamperometry (CA), were performed in a three-electrode cell. The three electrodes were the working electrode (tungsten wire, r = 0.5 mm), counter electrode (platinum plate, 10 × 10 mm^2^) and reference electrode (silver wire, r = 0.5 mm). Before every electrochemical measurement, the working electrode was polished with metallographic sandpaper and then polished with Al_2_O_3_ polishing powder with a particle size of 0.5 μm until the mirror surface became bright. Next, the measuring electrodes were ultrasonically cleaned with distilled water for 15 min. Finally, they were dried at 323 K under vacuum.

Cyclic voltammetry (CV) and chronoamperometry (CA) were performed with a potentiostat-galvanostat electrochemical instrument (Solartron 1287, Farnborough, UK). The potentiostat potential range was from −0.8 V to 0.8 V, and the sweep rate was from 20 mV·s^−1^ to 50 mV·s^−1^.

Bi films were prepared in ChCl-MA-BiCl_3_ by electrodeposition experiments. The electrodeposition experiment used a tungsten substrate (10 × 10 mm^2^) to obtain deposits. After the experiment, the deposits were washed with anhydrous ethanol to remove the electrolyte adhered to the electrode surface, and then dried under vacuum. Figure 1 is the schematic representation of the electrochemistry and electrodeposition experiment apparatus.

### 2.5. Physicochemical Characterization

The surface morphology of the deposits was determined by scanning electron microscopy (SEM, Zeiss Ultra, Oberkochen, Germany) with a 3–6 kV accelerating voltage. The deposit composition and chemical state were determined using X-ray photoelectron spectroscopy (XPS, Escalab 250, Thermo VG Scientific Ltd., West Sussex, UK) with an Al filament emitting X-rays at 1486.6 eV.

## 3. Results and Discussion

### 3.1. FTIR Spectra and Raman Spectroscopy Analysis

The FTIR spectra of ChCl-MA and the electrolyte containing different concentrations of Bi(III) are shown in Figure 2. The broad band appears at 3328 cm^−1^, indicating the formation of hydrogen bonds between ChCl and malonic acid, such as H−O···H, N−H···O and O−H···Cl [14]. The band at 3028 cm^−1^ can be attributed to the O−H stretching vibration in the malonic acid structure. The band at 1723 cm^−1^ is attributed to a carbonyl compound and C=O stretching vibration. Bougeard et al. reported that the broad peak reflects two dimer rings with almost equal hydrogen strength, which may be related to the two identical −C=O groups of malonic acid [15]. The band at 1482 cm^−1^ is attributed to the CH_2_ bending of an alkyl group, which is the prominent group in all ChCl-based DESs, 1162 cm^−1^ to the C-O stretching of an aliphatic ketone group and 872 cm^−1^ to a C-N^+^ symmetric stretching [16,17]. The band at 955 cm^−1^ is the asymmetric stretching vibration of the −CCO bond, which is basically the same as the corresponding peak of free ChCl, thus indicating the existence of a Ch^+^ structure in DES. There was no change in the infrared spectrum when BiCl_3_ was added to the ChCl-MA, which showed that the structure of Ch^+^ was not destroyed in the ChCl-MA.

Figure 3 shows the Raman spectra of ChCl-MA and the electrolyte containing different concentrations of Bi(III). The sharper band at 719 cm^−1^ is attributed to the symmetrical stretching vibration of the four C-N bonds (ν_1_) in the choline group. The bands at 865 cm^−1^ and 954 cm^−1^ are attributed to the symmetric (ν_2_) and asymmetric (ν_3_,ν_4_) stretching vibrations of the C–N bond, respectively. The weaker band at 768 cm^−1^ is due to the symmetrical tensile vibration of the four C–N bonds (ν_1_) in the trans-configuration of the choline group O–C–C–N^+^ backbone [18]. The band at 1437 cm^−1^ is due to the bending vibration of the –OH bond. The bands at 2961 cm^−1^ and 3031 cm^−1^ are caused by the stretching vibration of C–H [19]. The band at 1728 cm^−1^ is attributed to the stretching vibration of the C=O bond in the –COOH group [20].

By comparing with the Raman spectrum of the ChCl-MA system, after adding BiCl_3_, a new band can be observed at 257 cm^−1^, as shown in the inset of Figure 3. Vieira et al. also found a new band at 251 cm^−1^ in the choline chloride-glycol system containing [NnBu_4_] [BiCl_4_], which was inferred to be caused by [BiCl_6_]^3−^ [21].

Oertel et al. analyzed the structure of Bi(III) to form chlorine-containing complexes in aqueous solution by Raman spectrometry. The results show that the main species was [BiCl_6_]^3−^ with a band at 263 cm^−1^ [22]. Haight Jr et al. reported that [BiCl_4_]^−^ and [BiCl_6_]^3−^ exist as chlorobismuth complexes in the acid-containing aqueous solutions, and a single equilibrium [Equation (1)] exists in the solution [23]:(1)$${\left[{\mathrm{BiCl}}_{4}\right]}^{-}+2{\mathrm{Cl}}^{-}={\left[{\mathrm{BiCl}}_{6}\right]}^{-}$$

Kenney et al. reported that the main species are [BiCl_4_]^−^, [BiCl_5_]^2−^ and [BiCl_6_]^3−^ in LiCl-KCl-BiCl_3_ molten salt by Raman spectroscopy [24]. Fung et al. studied the major species in BiCl_3_-KCl molten mixtures by Raman spectroscopy. They speculated that the molten mixture of BiCl_3_-KCl formed anionic species of [BiCl_4_]^−^ and [BiCl_6_]^3−^ structures [25]. Although there is still some uncertainty about the presence of [BiCl_5_]^2−^, the equilibrium in chloride-containing solutions [Equation (1)] is accepted by many researchers. Therefore, it is concluded that Bi(III) also forms [BiCl_6_]^3−^ with chloride in ChCl-MA-BiCl_3_ by the complexation reaction. 

### 3.2. Viscosity and Conductivity Analysis

The relationship between the viscosity and temperature of ChCl-MA containing different BiCl_3_ concentrations is shown in Figure 4. The viscosity of ChCl-MA-BiCl_3_ ranging from 200 to 1200 mPa·s at temperatures from 363 K to 323 K is higher than that of most conventional solvents but similar to that of DES [26]. This is because the viscosity of the fluid is mainly caused by the internal friction between the molecules. The eutectic solvent of choline chloride is composed of large-size ions and small-volume holes. The molecules are mainly connected by hydrogen bonds. A large number of hydrogen-bond network structures decrease the mobility of free components in DES [27,28]. The viscosity of DES is mainly determined by van der Waals forces and hydrogen bonds. When the temperature increases, the van der Waals forces between the different groups in the DES begin to weaken, which leads to a decrease in viscosity. Halide salts can also form complexes with hydrogen bond honors. When BiCl_3_ is added, the number of molecules per unit volume increases, which causes the van der Waals forces between different groups to become stronger. As a result, a network structure forms between different groups, which causes the viscosity to increase [29].

The relationship between viscosity and temperature can be described by the Arrhenius empirical equation:lnη = lnη_0_ + E_η_/RT,(2)
where η_0_ is a constant, R is the gas constant, T is the absolute temperature and E_η_ is the energy for the activation of viscous flow.

Figure 5 shows the relationship diagram between ln_η_ and T^−1^, which is a linear relationship. Equation (2) can be used to describe the effect of temperature on electrolyte viscosity. The calculation results of E_η_ are shown in Table 1. E_η_ increases with the increasing BiCl_3_ concentration. As the concentration increases from 0.01 to 0.1 mol·L^−1^, E_η_ increases from 25.88 to 35.62 kJ·mol^−1^. Abbott et al. suggested that the energy for activation of viscous flow, E_η_, is determined by the ion-to-hole ratio; the higher the ion-to-hole ratio, the greater the value of E_η_. With the further increase in the bismuth salt content in DES, the reduction of the pore size becomes the main factor, which causes the E_η_ value to start to increase [11].

Figure 6 shows the conductivity of ChCl-MA containing different BiCl_3_ concentrations. The conductivity increases with the increasing temperature. The conductivity of 0.01 M Bi(III) in ChCl–MA is 3.24 ms·cm^−1^ at 363 K. This is because the temperature of the system increases and the viscosity of the electrolyte begins to decrease. At this time, the fluidity of the ions in the electrolyte increases, and the resistance of the ion movement decreases, which facilitates the migration of ions, which leads to the conductivity of the electrolyte system gradually increasing. It can also be seen that the conductivity of the electrolyte decreases with increasing bismuth salt concentration. Kityk et al. studied the influence of Ni(II) salt concentration on viscosity and conductivity within a certain range in a eutectic solvent system composed of choline chloride and ethylene glycol using NiCl_2_·6H_2_O as the nickel source. The results showed that the conductivity was controlled by ion mobility rather than the concentration of charge carriers [30]. The conductivity is not determined by the concentration of ions, but is mainly determined by the availability of suitably sized gaps. When the concentration of BiCl_3_ added to the electrolyte gradually increases, the conductivity value gradually decreases. This behavior may be due to the complexation behavior of Bi(III) ions in DES and the change in total electrolyte viscosity.

The relationship between conductivity and temperature can be described by the Arrhenius empirical equation:Lnк = lnк_0_ + E_к_/(RT),(3)
where к_0_ is a constant, R is the gas constant, E_к_ is the activation energy of electrical conductivity and T is the absolute temperature.

Figure 7 shows the relationship diagram between ln_к_ and T^−1^; it can be seen that it is a linear relationship. The calculation results of E_к_ are shown in Table 2. With the increase in BiCl_3_ concentration, the activation energy of electrical conductivity gradually increases, and its value increases from 24.63 to 30.32 kJ·mol^−1^. By comparing Table 1 and Table 2, it can be seen that the energy for activation of viscous flow is higher than the activation energy of electrical conductivity in the same electrolyte system, which is a typical phenomenon in various ionic liquids. 

### 3.3. Cyclic Voltammetry

The electrochemical windows of ChCl-MA were determined using cyclic voltammetry (Figure 8). The cyclic voltammograms of ChCl-MA-BiCl_3_ showed that only one pair of redox peaks was observed. The cathodic current peak observed at −0.36 V is attributed to the reduction of Bi(III), and an anodic current peak observed at 0.04 V is due to the stripping of the Bi.

A cathodic current crossover loop occurred during the reversal sweep, which indicated that the reduction of Bi(III) in ChCl-MA is a nucleation and growth process driven by an overpotential and features related to nucleation, followed by diffusion to limit the growth process. This phenomenon is similar to that reported by Sakita et al. [31].

#### 3.3.1. The Effect of Scanning Rate

Figure 9 shows the CV curves of ChCl-MA containing 0.10 M Bi(III) at different scan rates at 343 K. It can be seen that the peak current density of the reduction peak and the oxidation peak increases and the reduction peak potential moves to the negative potential direction with the increase in the scan rate. Theoretically, the peak potential of the reversible electrode reaction does not change with the change in the scan rates. The potential difference between the cathodic peak potential and the half peak potential |*E_p_*−*E*_*p*/2_| increases with the increase in the scan rate. The minimum value of |*E_p_*−*E*_*p*/2_| (85 mV, 343 K) is higher than the theoretical value (23 mV, 343 K) for the reversible process. It can be concluded that the reduction of Bi(III) in ChCl-MA is a quasi-reversible reaction.

Figure 10 shows the relationship between the square root of the scan rate (*v*^1/2^) and the cathodic peak current density (j_*p*_). It can be seen that *v*^1/2^ has a good linear relationship with j_*p*_, which is consistent with the diffusion control process reported by Manh et al. [32]. These characteristics show that the reduction process of Bi(III) in ChCl-MA is a quasi-reversible reaction controlled by diffusion.

Equation (4) is the relationship between the cathodic transfer coefficient of the electrode reaction in the irreversible process and the reduction peak potential and half-peak potential difference, and Equation (5) is the diffusion coefficient calculation formula of the irreversible reaction [33]. They are also suitable for quasi-reversible reactions [34].
(4)$$\left|E_{p}-E_{p/2}\right|=1.857RT/\left(\alpha nF\right),$$
(5)$$j_{p}=0.4958nFc^{0}{\left(\alpha nF/RT\right)}^{1/2}D^{1/2}v^{1/2},$$
where *E_p_* is the cathode peak potential, *E_p_*_/2_ is the cathode half peak potential, *R* is the ideal gas constant (8.314), *n* is the electron transfer number, *F* is the Faraday constant, j_*p*_ is the cathode peak current density, *c*^0^ is the concentration of Bi(III), *v* is the scan rate and *α* and *D* are the transfer coefficient and the diffusion coefficient, respectively.

Calculated by Equation (4) at 343 K, the average transfer coefficient of Bi(III) is 0.125. The diffusion coefficient of Bi(III) is calculated by Equation (5) to be 1.84 × 10^−9^ cm^2^·s^−1^. The values of Bi(III) in ChCl-ethylene glycol-[NnBu_4_][BiCl_4_] and AlCl_3_/N-(n-butyl)pyridinium chloride melt are 8.74 × 10^−8^ cm^2^·s^−1^ and 2.0 × 10^−7^ cm^2^·s^−1^, respectively, and the calculated values in this study are smaller [35], which may be due to the difference in viscosity of DESs, resulting in a certain difference in the diffusion coefficients.

#### 3.3.2. The Effect of Temperature

The CV curves of the 0.10 M ChCl-MA-BiCl_3_ electrolyte at different temperatures are shown in Figure 11. It can be seen that when the temperature increases, the peak current density of the oxidation peak and the reduction peak gradually increase, and the reduction peak potential gradually shifts to the positive potential direction (inset of Figure 10), which indicates that the rate of the Bi(III) reduction reaction increases with increasing temperature. On the one hand, the viscosity of the electrolyte decreases due to the increase in temperature, and thus, the higher mass transfer rate is obtained, which improves the diffusion efficiency of Bi(III). On the other hand, as the temperature increases, the thermal motion of the electroactive substance and the migration energy of Bi(III) increase. This causes the reduction potential of Bi(III) ions to be less affected by polarization, and the reduction peak shifts to the positive potential [36]. 

#### 3.3.3. The Effect of Bismuth Chloride Concentrations

Figure 12 shows the CV curve with different concentrations of BiCl_3_ in ChCl-MA at 343 K. It can be seen that the current density of the oxidation peak and reduction peak increases, and the potential of the reduction peak shifts toward the positive potential with the increase in Bi(III) ion concentration, which indicates that the presence of a large number of electroactive species at higher concentrations and the bismuth cathodic deposition are diffusion-controlled processes [37]. 

### 3.4. Chronoamperometry

Chronoamperometry was used to study the nucleation mechanism of Bi electrodeposition. The typical current-time transient curve is shown in Figure 13. It can be seen that the current-time transient curve presents three stages. The first part is the sudden drop in current corresponding to the current decay during the charging of the electric double layer [38]. The current then gradually rises corresponding to the formation and growth of Bi nuclei. In the latter region, the current value increases due to the higher Bi nucleation density, and the time required to reach the maximum current decreases as the applied potential increases [39]. In the third part, the current gradually decays from the maximum value to a stable value corresponding to the formation of a new diffusion layer, further indicating that the reduction of Bi(III) in ChCl-MA is controlled by diffusion [40].

According to the Scharifker–Hills nucleation model, the nucleation/growth mechanism of the metal deposition process includes instantaneous nucleation (IN) and progressive nucleation (PN) [41]. The (*j*/*j_m_*)^2^ − *t*/*t_m_* diagram of theoretical instantaneous nucleation and progressive nucleation can be described according to Equations (6) and (7). The experimental data obtained by the chronoamperometry were also subjected to the same dimensionless processing, and the obtained curve is shown in Figure 13.
(6)$$\mathrm{Instantaneous}\;\mathrm{nucleation}:\;\frac{j^{2}}{j_{m}^{2}}=\frac{1.9542}{t/t_{m}}{\left[1-\exp\left(-1.2564\frac{t}{t_{m}}\right)\right]}^{2}$$
(7)$$\mathrm{Progressive}\;\mathrm{nucleation}:\;\frac{j^{2}}{j_{m}^{2}}=\frac{1.2254}{t/t_{m}}{\left\{1-\exp\left[-2.3367{\left(t/t_{m}\right)}^{2}\right]\right\}}^{2}$$

It can be seen from Figure 14 that the dimensionless curve gradually approaches the instantaneous nucleation curve as the applied deposition potential becomes negative, which indicates that when the deposition potential is applied, the instantaneously active sites on the surface of the substrate are occupied by bismuth nuclei, and no more bismuth nuclei will form with prolonged time [42]. It is concluded that the electrodeposition nucleation mode of Bi in the ChCl-MA system at 343 K followed the three-dimensional instantaneous nucleation controlled by diffusion. When the applied potential is −0.55 V, the experimental curves are in the region of the three-dimensional nucleation PN and IN when t/t_m_ ≤ 1, which is related to the existence of reactions concomitant with the nucleation and growth mechanisms, such as protons or residual water reduction [43,44]. 

### 3.5. Electrodeposition and Characterization of Bi

Figure 15 shows the surface images of the bismuth electrodeposited on the tungsten substrate in ChCl-MA-BiCl_3_ for 600 s at different cathodic potentials (−0.47 V, −0.55 V). Figure 15a shows that when the deposition voltage is −0.47 V, the particle size of the obtained bismuth metal is approximately 100 nm, and its microscopic morphology is a prismatic shape. As the potential becomes more negative (−0.55 V), it can be clearly observed from Figure 15b that the grain size increases and the film is denser. This is because the application of the deposition voltage is the driving force for the formation of the Bi nucleus, and the negative potential causes the growth rate of the nucleus to increase, which accelerates the growth of the Bi nucleus.

Figure 16 shows the XPS spectrum of the electrodeposition product obtained at 323 K with a deposition potential of −0.55 V for 600 s. The peaks at 158.6 eV and 163.9 eV correspond to the binding energy of Bi 4f_7/2_ and Bi 4f_5/2_, respectively. Two contributions are fitted: the Bi 4f_7/2_ binding energies of 156.8 eV and 158.6 eV, and the Bi 4f_5/2_ binding energies of 162.3 eV and 163.9 eV, which correspond to the metallic bismuth and the trivalent oxidation state of bismuth, respectively [45,46]. Based on XPS analysis, metallic bismuth and bismuth oxide (Bi_2_O_3_) are present in the deposit.

## 4. Conclusions

Bismuth was successfully prepared from ChCl-MA eutectic solvent by the electrodeposition method. CV results show that the electrodeposition of Bi(III) is a diffusion-controlled quasi-reversible reaction in the ChCl-MA system, and its cyclic voltammetry behavior is affected by temperature and concentration. The FTIR spectrum shows the presence of hydrogen bonds in the electrolyte. Raman spectroscopy shows that Bi(III) forms a [BiCl_6_]^3−^ complex in ChCl-MA. The viscosity and conductivity of ChCl-MA-BiCl_3_ are greatly affected by the temperature, which obeys classical Arrhenius behavior. The viscosity and conductivity of the electrolyte system have a significant effect on the diffusion coefficient of Bi(III) in ChCl-MA. Chronoamperometry indicated that the nucleation mode of bismuth on the tungsten electrode at 343 K is three-dimensional instantaneous nucleation. The SEM image shows that the bismuth film can be obtained from ChCl-MA by electrodeposition, and the morphology of the deposit is affected by the deposition potential. XPS results show that bismuth and bismuth oxide are present in the electrodeposited product.

## Figures and Tables

**Figure 1 materials-16-00415-f001:**
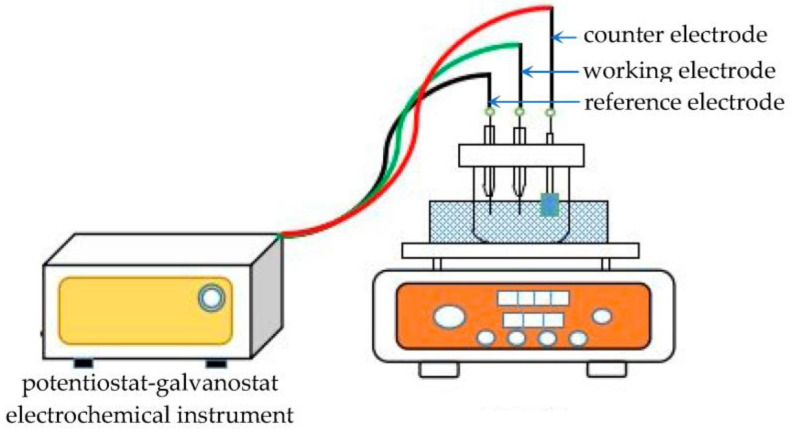
Schematic representation of the electrochemistry and electrodeposition experiment apparatus.

**Figure 2 materials-16-00415-f002:**
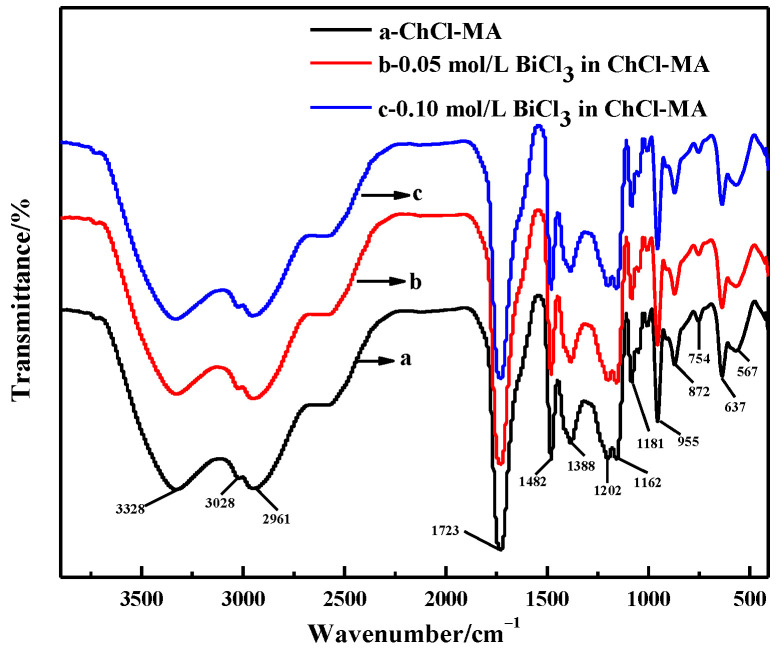
FTIR spectra of ChCl-MA system and ChCl-MA-BiCl_3_.

**Figure 3 materials-16-00415-f003:**
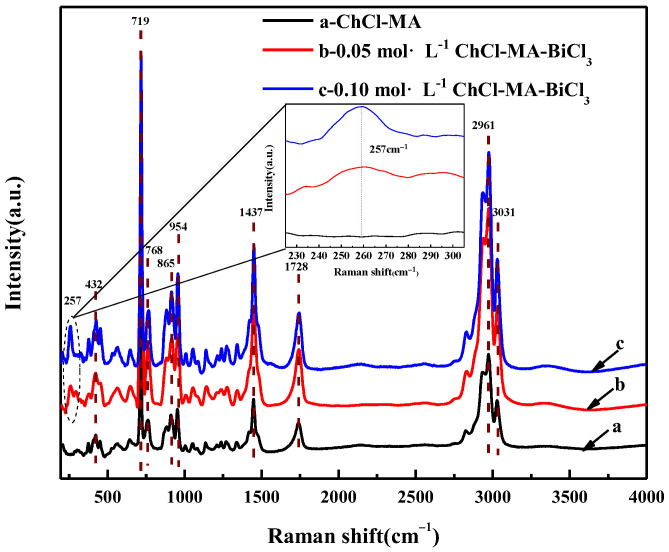
Raman spectra of ChCl-MA system and ChCl-MA-BiCl_3_.

**Figure 4 materials-16-00415-f004:**
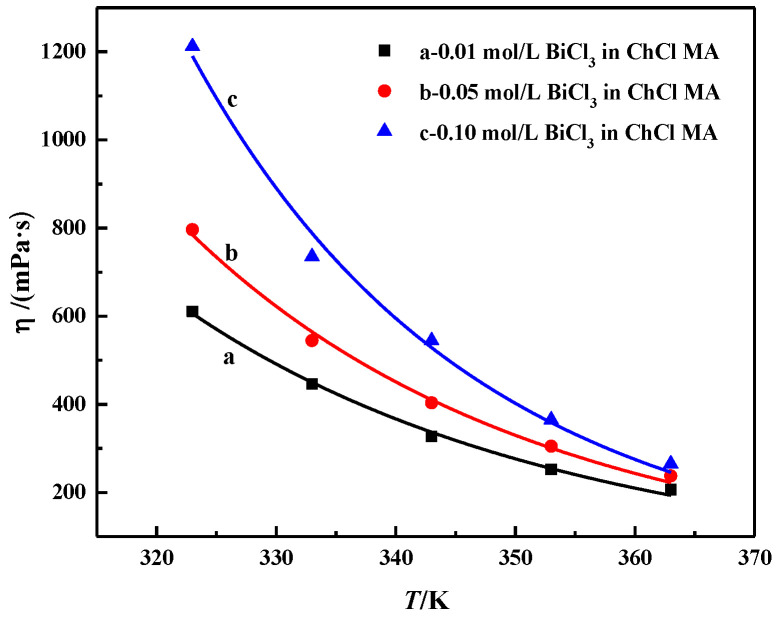
The relationship between the viscosity of the ChCl-MA-BiCl_3_ system and the change of BiCl_3_ concentration with temperature.

**Figure 5 materials-16-00415-f005:**
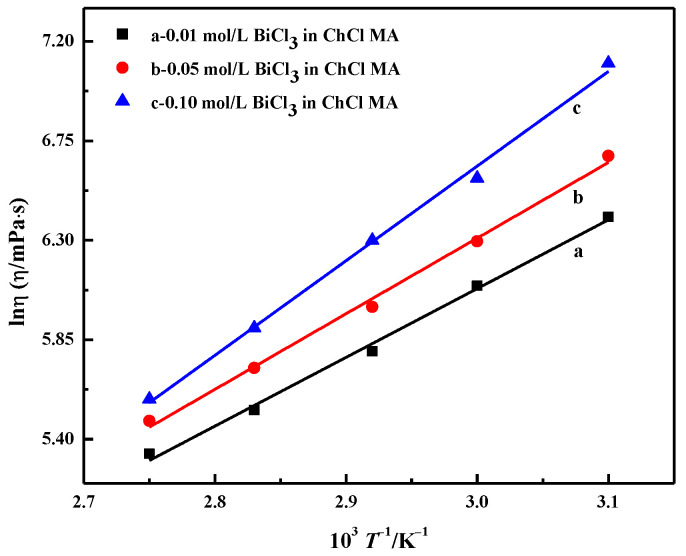
Relationship between the natural logarithm of viscosity and temperature reciprocal of ChCl-MA containing different concentrations of BiCl_3_.

**Figure 6 materials-16-00415-f006:**
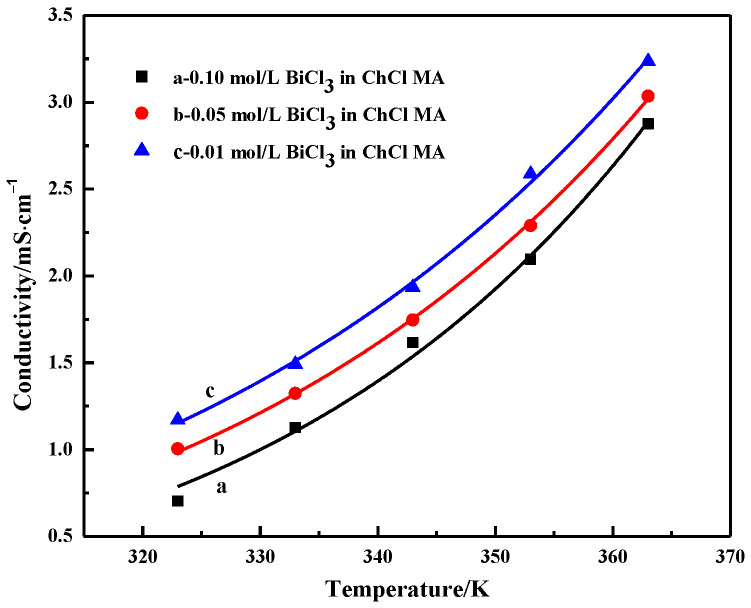
Conductivity of ChCl-MA-BiCl_3_ as a function of temperature and concentration of BiCl_3_.

**Figure 7 materials-16-00415-f007:**
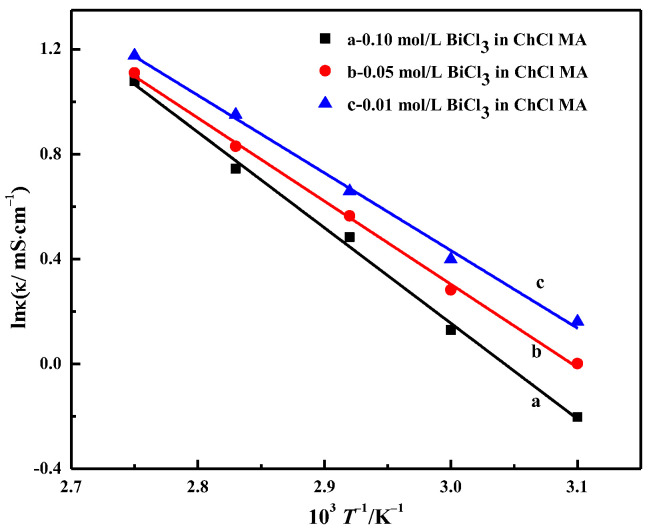
Relationship between the natural logarithm of conductivity and temperature reciprocal of ChCl-MA containing different concentrations of BiCl_3_.

**Figure 8 materials-16-00415-f008:**
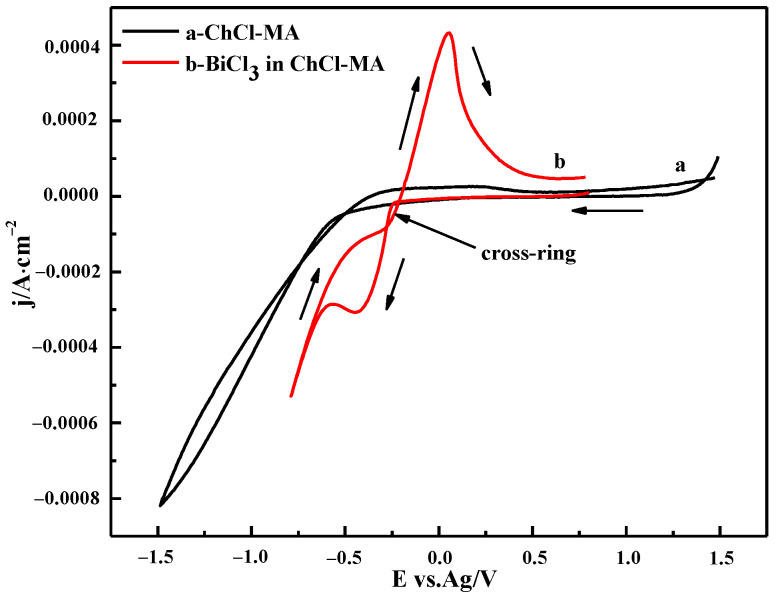
Cyclic voltammogram of the ChCl-MA blank system and ChCl-MA-BiCl_3_ (T = 343 K, v = 20 mV·s^−1^, c(BiCl_3_) = 0.10 M).

**Figure 9 materials-16-00415-f009:**
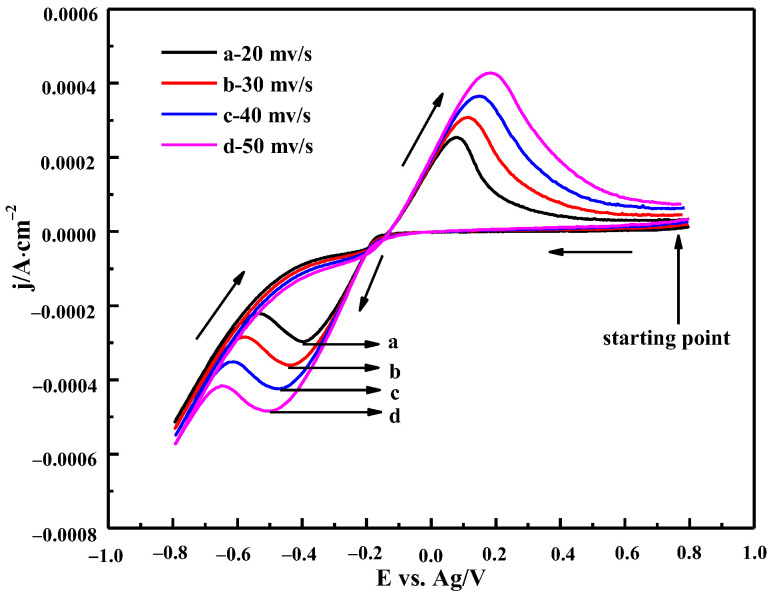
Cyclic voltammetry curves of ChCl-MA-BiCl_3_ at different scan rates (T = 343 K, c(BiCl_3_)) = 0.10 M).

**Figure 10 materials-16-00415-f010:**
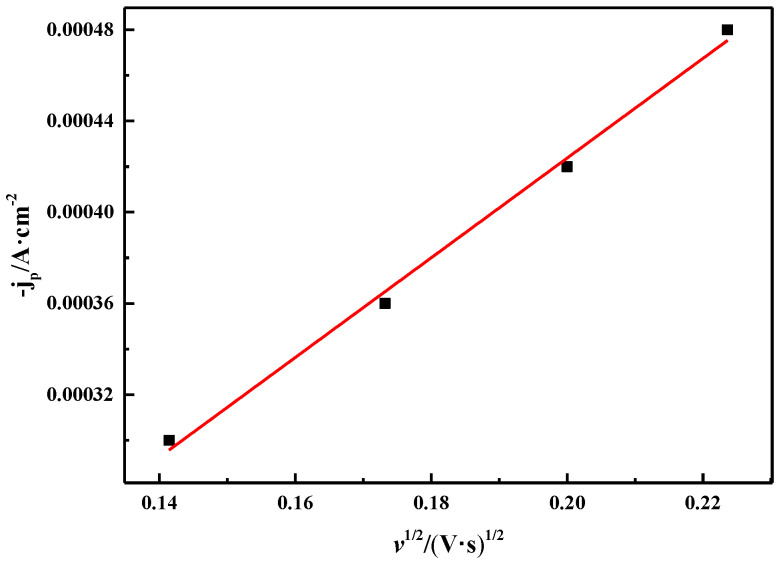
Relationship between square root of different scan rates and corresponding cathodic peak current densities (T = 343 K, c(BiCl_3_) = 0.10 M).

**Figure 11 materials-16-00415-f011:**
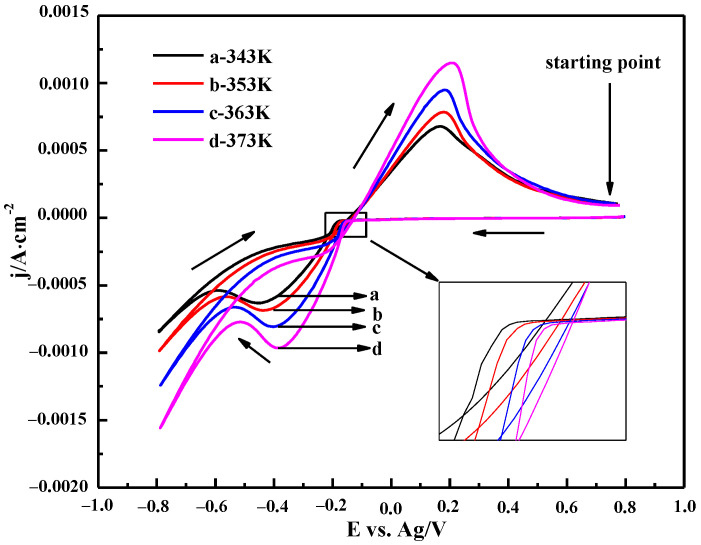
Cyclic voltammetry curves of ChCl-MA-BiCl_3_ at different temperatures (v = 50 mV·s^−1^, c(BiCl_3_) = 0.10 M).

**Figure 12 materials-16-00415-f012:**
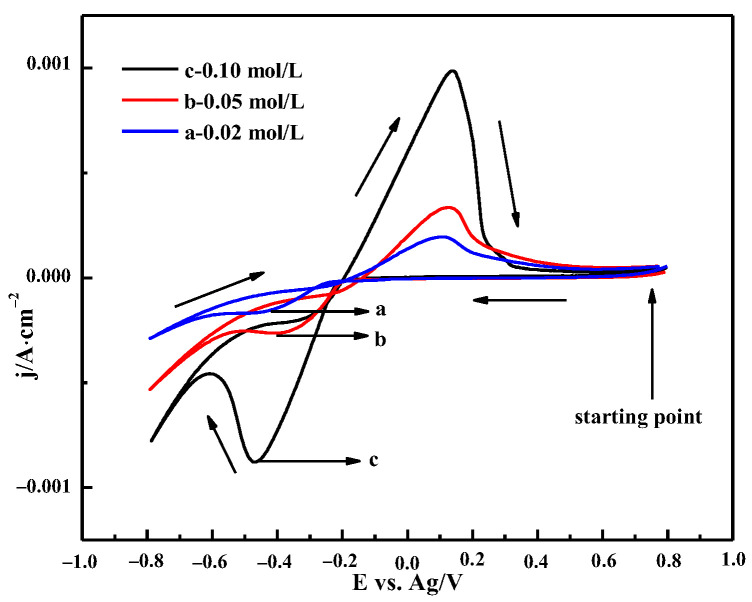
Cyclic voltammetry curves of ChCl-MA-BiCl_3_ at different concentrations (v = 50 mV·s^−1^, T = 343 K).

**Figure 13 materials-16-00415-f013:**
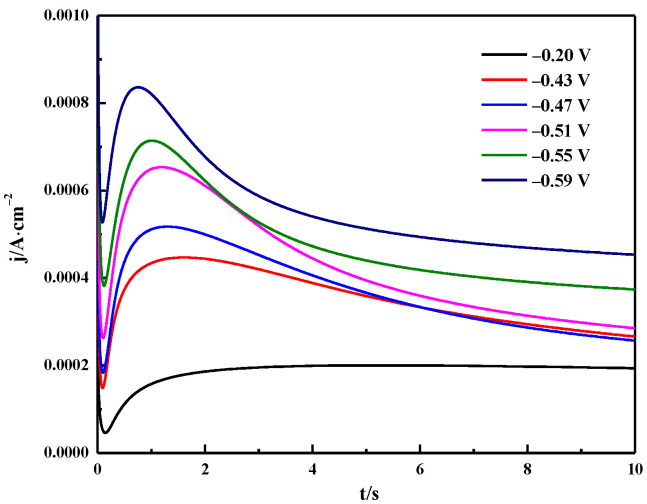
Potentiostatic current transients obtained in the Bi electrodeposition process on the tungsten electrode in ChCl-MA-BiCl_3_ at different potentials (T =343 K, c(BiCl_3_) = 0.10 M).

**Figure 14 materials-16-00415-f014:**
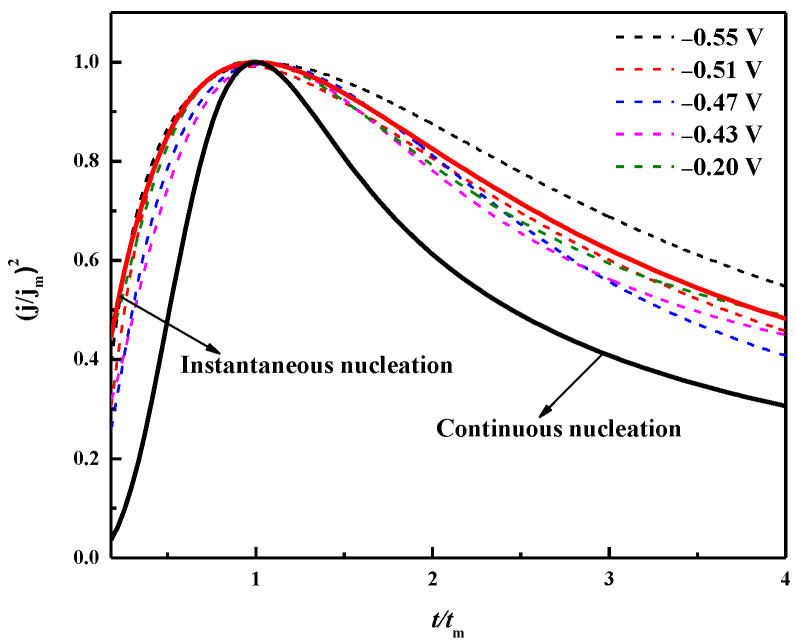
Chronoamperometric experimental data dimension curve and theoretical three-dimensional instantaneous/continuous comparison of nucleation model curves.

**Figure 15 materials-16-00415-f015:**
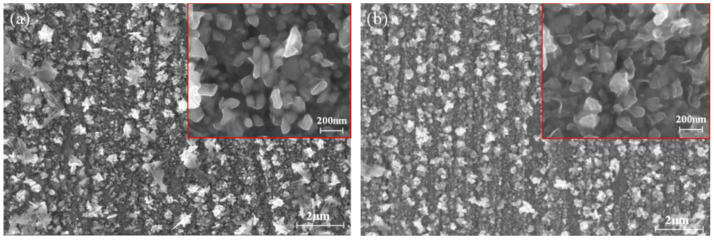
SEM images of bismuth electrodeposition products obtained under different deposition potentials: (**a**) −0.47 V, (**b**) −0.55 V (T = 323 K, c(BiCl_3_) = 0.10 M).

**Figure 16 materials-16-00415-f016:**
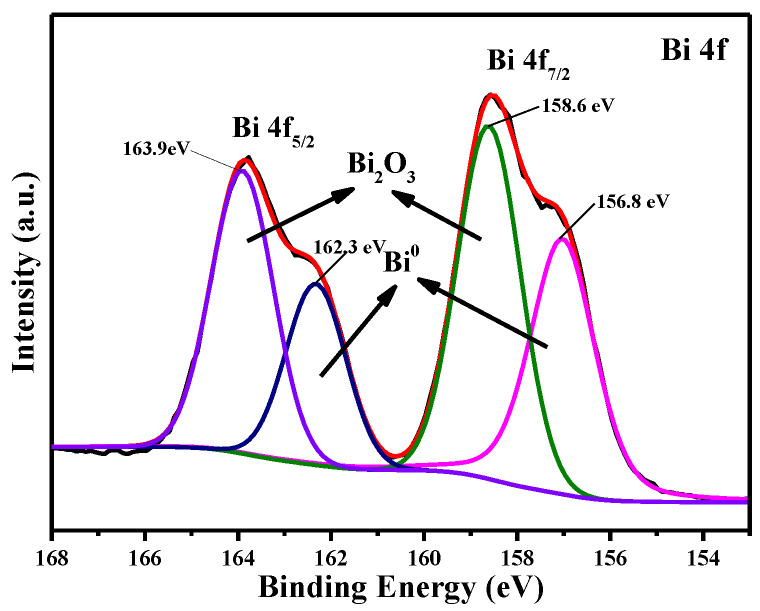
The XPS spectra of the electrodeposition product obtained at −0.55 V (T = 323 K, c(BiCl_3_) = 0.10 M).

**Table 1 materials-16-00415-t001:** Arrhenius Equations of Viscosity Along with the Corresponding Parameters.

Concentration/(mol∙L^−1^)	Linear Fitting Equation	R^2^	η_0_	E_η_/(kJ∙mol^−1^)
0.01	ln_η_ = −3.2581 + 3113.18 T^−1^	0.9946	3.85 × 10^−2^	25.88
0.05	ln_η_ = −3.9617 + 3423.89 T^−1^	0.9949	1.90 × 10^−2^	28.47
0.10	ln_η_ = −6.2160 + 4284.04 T^−1^	0.9956	2.00 × 10^−3^	35.62

**Table 2 materials-16-00415-t002:** Arrhenius Equations of Conductivity Along with the Corresponding Parameters.

Concentration/(mol∙L^−1^)	Linear Fitting Equation	*R* ^2^	*к* _0_	*E_к_*/(kJ∙mol^−1^)
0.01	Lnк = 9.3199 − 2962.47 *T*^−1^	0.9957	1.12 × 10^4^	24.63
0.05	Lnк = 9.8275 − 3174.62 *T*^−1^	0.9981	1.85 × 10^4^	26.39
0.10	Lnк = 11.0967 − 3647.27 *T*^−1^	0.9958	6.60 × 10^4^	30.32

## Data Availability

Not applicable.

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
