# Peer review of "Electrodeposition of Bi from Choline Chloride-Malonic Acid Deep Eutectic Solvent"

_materials, 2023, doi:10.3390/ma16010415_

Round 1
Reviewer 1 Report
An interesting research paper.
Author Response
Dear reviews,
Thanks for your review.
Reviewer 2 Report
1. There are some grammatical and misspelling errors. For instance, the next sentence: “The higher temperature of electrolysis leads to high consumption of power, which is mainly because of the high liquidus temperature of the molten salt electrolyte”
2. What about the thermal stability of the DES? Delgado et. al, which is cited in this work regarding the FTIR spectra, shows a deep analysis of thermal stability of several DESs, including that presented here.
3. I think that the purpose of the paper is not clear. I miss a comparison with other DES or other matrixes to check the performance of ChCl:MA as a good solvent for the electrodeposition of Bi. At the end of the day, I cannot figure out if this method enhances current technologies or not.
Author Response
Dear reviewer,
Thanks for your review.
The responds to your comments are in the attachment.

Reviewer 3 Report
Title- Electrodeposition of Bi from Choline Chloride-Malonic Acid Deep Eutectic Solvent
Authors- X. Cao, H. Wang, T. Liu, Y. Shi, X. Xue
Manuscript Id- Materials-2046549
In this manuscript, the authors have performed the deposition of Bi on a tungsten substrate by electrodeposition means. FTIR, Raman, and XPS are used to characterize the nature of electrodeposited products. Furthermore, the performance of the synthesized film was probed by electrochemical characterization. After a careful review with interest, the reviewer has below comments that needs to be incorporated extensively in the manuscript. This manuscript also falls short on the grounds of scientific writing and the English language to meet the standards of MDPI Materials. The authors are advised to revise the manuscript extensively, and hence recommend a major revision.
Below are some comments/suggestions for the authors;
[1] Authors should consider rewriting of abstract part in a more scientific manner. It would be appreciated that authors should also add some factual data in the abstract for better and clear readability. For ex., how much deposition potential, %electrochemical products, and electrochemical constant values?
[2] Although not new but electrochemical deposition is a hot topic, it would be nice to incorporate its schematic containing electrode assembly. Also, authors need to verify whether its electrodeposition or electrochemical deposition. I guess, its electrochemical deposition. Please look into the articles from Annexure 1 (uploaded separately).
[3] It would be appreciated that authors should also discuss some pros and cons of electrochemical deposition process in the introduction section. They should also clearly mention how it is advantageous over conventional chemical treatments. This will eventually help to define the motivation and novelty of the manuscript. Articles enclosed in Annexure 1 are important and may help in this regard.
[4] Authors also need to work on introduction section extensively. They failed to build a story and subsequently the novelty.
[5] A quick question: what was the laser spot size in Raman experiments? What was the magnification of the lens? How much integration time? What was the grit size? Needs more clarification.
[6] Again, during conductivity and viscosity measurement, how much was the resolution of both the instruments used for studying the above properties? Please explain in detail.
[7] In Section 2; Materials and Methods; please give a detailed description of electrochemical experiments. For ex. What is the resolution of the potentiostat? How much was the sweep rate? Please have a look at the articles from Annexure 1 and how these authors have described the experimental section.
[8] Are the % of constituents used in electrolyte preparation optimized? Did the authors perform some trial experiments? How do they arrive at determining levels? A little explanation of this would be appreciated.
[9] In Fig. 1, peaks of all the samples are at the same wavenumbers. Any comments? Did authors observe peak shifting? The transmittance values might be the same or different. Again, does the transmittance axis indicate relative transmittance (say T/Tmax)?
[10] A quick question: how did the authors fit the Raman spectra? Which fitting parameters they used and why? Needs more clarification.
[11] Is the peak shift less than 2 or 3 cm-1 significant to arrive at conclusion? How many Raman spectra did the authors perform on each sample? Are the results reproducible? Please address this positively along with comment no. 5.
[12] In Fig. 3, the difference in viscosity is significant at a lower temp. but as the temp. increases, all the samples exhibit similar viscosity values. Any comments? Again, please mention the viscosity values with standard deviation.
[13] The reviewer is curious about the “potential difference between cathodic potential and half potential increases with increase in scan rate”. Please explain why.
[14] Furthermore, thin films are more prone to porosity. It would be highly appreciated that authors should also take porosity factor into consideration and enrich the discussion part. For ex. it would be apt to state the effect of porosity on the electrochemical properties of film or electrochemical product formation. Articles from Annexure 1 are important and may help in this regard.
[15] It would be also interesting to see how the difference in viscosity leads to the diffusion constants and eventually electrochemical performance.
[16] The reviewer is also curious about the instantaneous and progressive nucleation and their subsequent effects on interfacial product formation.
[17] Last section 3.4 should go as section 3.1.
In my opinion;
[1] This reviewer appreciates this nice initiative for the electrochemical deposition of Bi in a deep eutectic solvent. However, manuscripts fall short from the discussion point of view. Authors are also advised to polish their manuscript for English language. Language is not scientific.
[2] The discussion in the manuscript is speculative. Authors are advised to look into the literature thoroughly and enrich the discussion part.
[3] Many unclear facts are presented in the manuscript. However, I advise authors to revise the manuscript extensively, and hence recommend a major revision.
Author Response
Dear reviewer,
Thanks for your review.
The main corrections in the paper and the responds to your comments are in the attachment.

Round 2
Reviewer 3 Report
Title- Electrodeposition of Bi from Choline Chloride-Malonic Acid Deep Eutectic Solvent
Authors- X. Cao, H. Wang, T. Liu, Y. Shi, X. Xue
Manuscript Id- Materials-2046549R1
The reviewer had a detailed look at the revised version of the manuscript. After a careful review, the reviewer is still not convinced with the revision made by the authors. There are many unclear facts that still exist in the manuscript. Surprisingly, the authors failed to address the comments raised by the reviewer. Hence, the reviewer cannot recommend the acceptance, and therefore advise authors to revise the manuscript extensively. Please look into the previous and present comments thoroughly, understand them properly, and consider these issues positively.
[1] The reviewer is not convinced with the address of comments 2, 3, and 4 (in round 1). Authors need to work extensively. They are advised to look into the articles from Annexure 1 (round 1) thoroughly. Articles mentioned in Annexure 1 are important but not cited in the previous version of the manuscript. This will help them to enrich the introduction and discussion part of the manuscript.
[2] Why does potential range from -0.8 V to +0.8 V? What is the rationale behind it? Please explain in detail.
[3] Regarding the address of comment no. 9 (round 1); if the structure is not changing then what is the cause of the different properties/behavior of ChCl-MA and BiCl3-ChCl-MA? Needs more justification.
[4] In the same vein, the author’s explanation for comments 9 and 11 (round 1) seems contradictory. In comment 9, they are arguing structure is not changing and in comment 11, they concluded that Raman shift might be due to structural change. Please have a detailed look into the literature and address both comments properly.
[5] Address for comment no. 12 seems superficial or speculative. At least cite some appropriate references for the argument.
[6] What is the quasi-reversible reaction? Please explain the mechanism in detail.
[7] Please give proper justification to comment no. 14? Currently, it seems that the authors are not interested in addressing the reviewer’s comments positively. Doing experiments in the future is fine but the reviewer expected a brief explanation (maybe in 3-4 lines) of the influence of the porosity factor. A similar would be the case for comments 15 and 16 (round 1).
In my opinion;
The authors failed to address the comments raised by the reviewer in the previous round 1. The suggestions raised by the reviewer during previous comments may be helpful but not given consideration. However, the above comments need to be amended mandatorily before the final publication of the manuscript. Therefore, a major revision is recommended again. Hope the authors will consider these comments and revise the manuscript positively.
Author Response
Dear reviewer,
We have response the comments, Please see the attachment.
Best regards,
Xiaozhou Cao

Round 3
Reviewer 3 Report
The authors still did not give a satisfactory explanation to the reviewer's comments. It seems they are not understanding the reviewer's suggestions properly. Nevertheless, I recommend the acceptance of the manuscript.